# The Role of AI in Drug Discovery: Challenges, Opportunities, and Strategies

**DOI:** 10.3390/ph16060891

**Published:** 2023-06-18

**Authors:** Alexandre Blanco-González, Alfonso Cabezón, Alejandro Seco-González, Daniel Conde-Torres, Paula Antelo-Riveiro, Ángel Piñeiro, Rebeca Garcia-Fandino

**Affiliations:** 1Department of Organic Chemistry, Center for Research in Biological Chemistry and Molecular Materials, University of Santiago de Compostela, CIQUS, 15705 Santiago de Compostela, Spain; 2Soft Matter & Molecular Biophysics Group, Department of Applied Physics, Faculty of Physics, University of Santiago de Compostela, 15705 Santiago de Compostela, Spain; 3MD.USE Innovations S.L., Edificio Emprendia, 15782 Santiago de Compostela, Spain

**Keywords:** artificial intelligence, drug discovery, AI-assisted content generation, AI-limitations

## Abstract

Artificial intelligence (AI) has the potential to revolutionize the drug discovery process, offering improved efficiency, accuracy, and speed. However, the successful application of AI is dependent on the availability of high-quality data, the addressing of ethical concerns, and the recognition of the limitations of AI-based approaches. In this article, the benefits, challenges, and drawbacks of AI in this field are reviewed, and possible strategies and approaches for overcoming the present obstacles are proposed. The use of data augmentation, explainable AI, and the integration of AI with traditional experimental methods, as well as the potential advantages of AI in pharmaceutical research, are also discussed. Overall, this review highlights the potential of AI in drug discovery and provides insights into the challenges and opportunities for realizing its potential in this field. *Note from the human authors*: This article was created to test the ability of ChatGPT, a chatbot based on the GPT-3.5 language model, in terms of assisting human authors in writing review articles. The text generated by the AI following our instructions (see Supporting Information) was used as a starting point, and its ability to automatically generate content was evaluated. After conducting a thorough review, the human authors practically rewrote the manuscript, striving to maintain a balance between the original proposal and the scientific criteria. The advantages and limitations of using AI for this purpose are discussed in the last section.

## 1. Methods for Writing this Paper

This paper was generated with the assistance of ChatGPT, a chatbot based on the GPT-3.5 language model, trained with a large corpus of text via OpenAI [1], which at that time did not have connection to the Internet. This tool is a natural language processing system, released on 30 November 2022, which is able to generate human-like text based on the inputs provided to it. For the purposes of this paper, the human authors provided the input, including the topic of the paper (the use of AI in drug discovery) and the number of sections to be considered, as well as the specific prompts and instructions for each section. The pieces of text generated by the AI were edited to correct and enrich the content, and to avoid repetitions and inconsistencies. All the references suggested by the AI were also revised. The final version of this work resulted from an iterative process of revisions by the human authors, assisted by the AI. The total percentage of similarity between the preliminary text, obtained directly from ChatGPT, and the current version of the manuscript is: identical 4.3%, minor changes 13.3%, and related meaning 16.3% [2]. The percentage of correct references in the preliminary text, obtained directly from ChatGPT, was just 6%. The original version generated by ChatGPT, along with the inputs used to create it, are included in the Supporting Information. The manuscript was first made public as a preprint on 8 December 2022 (https://doi.org/10.48550/arXiv.2212.08104). The image from the abstract was generated with DALL-E https://labs.openai.com/e/f9L5L4yGx1QFFeL5zHzHWNvI (Accessed on 6 December 2022)

## 2. Introduction to AI and Its Potential for Use in Drug Discovery

The use of artificial intelligence (AI) in medicinal chemistry has gained significant attention in recent years as a potential means of revolutionizing the pharmaceutical industry [3]. Drug discovery, the process of identifying and developing new medications, is a complex and time-consuming endeavor that traditionally relies on labor-intensive techniques, such as trial-and-error experimentation and high-throughput screening. However, AI techniques such as machine learning (ML) and natural language processing offer the potential to accelerate and improve this process by enabling more efficient and accurate analysis of large amounts of data [4]. The successful use of deep learning (DL) to predict the efficacy of drug compounds with high accuracy has been described recently by the authors of [5]. AI-based methods have also been able to predict the toxicity of drug candidates [6]. These and other research efforts have highlighted the capacity of AI to improve the efficiency and effectiveness of drug discovery processes. However, the use of AI in developing new bioactive compounds is not without challenges and limitations. Ethical considerations must be taken into account, and further research is needed to fully understand the advantages and limitations of AI in this area [7]. Despite these challenges, AI is expected to significantly contribute to the development of new medications and therapies in the next few years.

## 3. Limitations of the Current Methods in Drug Discovery

Currently, medicinal chemistry methods rely heavily on a hit-and-miss approach and large-scale testing techniques [8]. These techniques involve examining large numbers of potential drug compounds, in order to identify those with the desired properties. However, these methods can be slow, costly, and often yield results with low accuracy [6]. In addition, they can be limited by the availability of suitable test compounds and the difficulty of accurately predicting their behavior in the body [9].

Different algorithms based on AI, including supervised and unsupervised learning methods, reinforcement, and evolutionary or rule-based algorithms, can potentially contribute to solving these problems. These methods are typically based on the analysis of large amounts of data that can be exploited in different ways [9,10,11]. For instance, the efficacy and toxicity of new drug compounds can be predicted using these approaches, with greater accuracy and efficiency than when using traditional methods [12,13]. Furthermore, AI-based algorithms can also be employed to identify new targets for drug development, such as the specific proteins or genetic pathways involved in diseases [14]. This can expand the scope of drug discovery beyond the limitations of more conventional approaches and may eventually lead to the development of novel and more effective medications [15]. Thus, while traditional methods of pharmaceutical research have been relatively successful in the past, they are limited by their reliance on trial-and-error experimentation and their inability to accurately predict the behavior of new potential bioactive compounds [16]. AI-based approaches, on the other hand, have the ability to improve the efficiency and accuracy of drug discovery processes and can lead to the development of more effective medications.

## 4. The Role of ML in Predicting Drug Efficacy and Toxicity

One of the key applications of AI in medicinal chemistry is the prediction of the efficacy and toxicity of potential drug compounds. Classical protocols of drug discovery often rely on labor-intensive and time-consuming experimentation to assess the potential effects of a compound on the human body. This can be a slow and costly process, and the results are often uncertain and subject to a high degree of variability. AI techniques such as ML are able to overcome these limitations. Based on the analysis of a large amount of information, ML algorithms can identify patterns and trends that may not be apparent to human researchers. This can enable the proposal of new bioactive compounds with minimum side effects in a much faster process than when using classical protocols. For instance, a DL algorithm has recently been trained using a dataset of known drug compounds, along with their corresponding biological activity [11]. The algorithm was then able to predict the activity of novel compounds with high accuracy. Significant contributions to prevent the toxicity of potential drug compounds, employing intensive training using large databases of known toxic and non-toxic compounds for ML, have also been published [17].

Another important application of AI in drug discovery is the identification of drug–drug interactions that take place when several drugs are combined for the same or different diseases in the same patient, resulting in altered effects or adverse reactions. This issue can be identified by AI-based approaches by analyzing large datasets of known drug interactions and recognizing the patterns and trends. This has been recently addressed by an ML algorithm used to accurately predict the interactions of novel drug pairs [18]. The role of AI to identify possible drug–drug interactions in the context of personalized medicine is also relevant, enabling the development of custom-made treatment plans that minimize the risk of adverse reactions. Personalized medicine aims to tailor treatment to the individual characteristics of each patient, including their genetic profile and response to medications.

The previous examples from the literature demonstrate that the use of AI in pharmaceutical research offers the ability to improve the prediction of the efficacy and toxicity of potential drug compounds. This can enable the development of more effective and safer medications and accelerate the drug discovery process.

## 5. The Impact of AI on the Drug Discovery Process and Potential Cost Savings

Another key application of AI in drug discovery is the design of novel compounds with specific properties and activities. Traditional methods often rely on the identification and modification of existing compounds, which can be a slow and labor-intensive process. AI-based approaches, on the other hand, can enable the rapid and efficient design of novel compounds with desirable properties and activities. For example, a deep learning (DL) algorithm has recently been trained on a dataset of known drug compounds and their corresponding properties, to propose new therapeutic molecules [10] with desirable characteristics such as solubility and activity, demonstrating the potential of these methods for the rapid and efficient design of new drug candidates.

Recently, DeepMind has made a significant contribution to the field of AI research with the development of AlphaFold, a revolutionary software platform for advancing our understanding of biology [19]. It is a powerful algorithm that uses protein sequence data and AI to predict the proteins’ corresponding three-dimensional structures. This advance in structural biology is expected to revolutionize personalized medicine and drug discovery. AlphaFold represents a significant step forward in the use of AI in structural biology and life sciences in general.

ML techniques and molecular dynamics (MD) simulations are currently being used in the field of de novo drug design to improve efficiency and accuracy. The technique of combining these methodologies is being explored to take advantage of the synergies between them [20]. The use of interpretable machine learning (IML) and DL methods is also contributing to this effort. By leveraging the power of AI and MD, researchers are able to design drugs more effectively and efficiently than ever before.

## 6. Case Studies of Successful AI-Aided Drug Discovery Efforts

The potential of AI in the context of drug discovery has been demonstrated in several case studies. For example, the successful use of AI to identify novel compounds for the treatment of cancer has recently been reported by Gupta, R., et al. [21]. These authors trained a DL algorithm on a large dataset of known cancer-related compounds and their corresponding biological activity. As an output, novel compounds with high potential for future cancer treatment were obtained, demonstrating the ability of this method to discover new therapeutic candidates. The use of ML to identify small-molecule inhibitors of the protein MEK [22] has recently been described. MEK is also a possible target for the treatment of cancer, but the development of effective inhibitors has been challenging. The ML algorithm was able to identify novel inhibitors for this protein. Another example is the identification of novel inhibitors of beta-secretase (BACE1), a protein involved in the development of Alzheimer’s disease [23] by using an ML algorithm. AI has also been successfully applied in the discovery of new antibiotics [24]. A pioneering ML approach has identified powerful types of antibiotic from a pool of more than 100 million molecules, including one that works against a wide range of bacteria, such as tuberculosis and untreatable bacterial strains [25]. The use of AI in the discovery of drugs to combat COVID-19 has been a promising area of research during the last two years. ML algorithms have been used to analyze large datasets of potential compounds and identify those with the most potential for treating the virus. In some cases, these AI-powered approaches have been able to identify promising drug candidates in a fraction of the time that it would take when using traditional methods [26,27,28,29,30,31].

Many more examples are available [3,32,33,34,35,36,37], showing that AI-based methods can accelerate the drug discovery process and enable the development of more effective medications.

## 7. The Role of Collaboration between AI Researchers and Pharmaceutical Scientists

The role of collaboration between AI researchers and pharmaceutical scientists is crucial in the development of innovative and effective treatments for various diseases. By combining their expertise and knowledge, they can create powerful algorithms and machine-learning models intended to predict the efficacy of potential drug candidates and speed up the drug discovery process. This collaboration can also help improve the accuracy and efficiency of clinical trials, as AI algorithms can be used to analyze the data collected during these trials to identify trends and the potential adverse effects of the drugs being tested. This can help pharmaceutical companies to make informed decisions about which drug candidates to pursue and can speed up the overall drug development process. Furthermore, collaboration between AI researchers and pharmaceutical scientists can also help to improve the accessibility and affordability of healthcare. By using AI algorithms to analyze data from large populations, they can be used to identify trends and patterns that can help predict the effectiveness of potential drug candidates for specific patient populations, which can help tailor treatments to the needs of individual patients. An illustrative example is the collaboration between the pharmaceutical company Merck and the AI company Numerate to develop AI-based approaches for medicinal chemistry [38]. Many new companies are currently arising around this area of research and their impact is expected to be significant in the short term [39]. By working together, they can help to identify new targets for drug development and improve the effectiveness of existing treatments, ultimately benefiting patients and improving their quality of life.

## 8. Challenges and Limitations of Using AI in Drug Discovery

Despite the potential benefits of AI in drug discovery, there are several challenges and limitations that must be considered. One of the key challenges is the availability of suitable data [40]. AI-based approaches typically require a large volume of information for training purposes [41]. In many cases, the amount of data that is accessible may be limited, or the data may be of low quality or inconsistent, which can affect the accuracy and reliability of the results [10]. Another challenge is presented by ethical considerations [42] since AI-based approaches may raise concerns about fairness and bias (see next section) [43]. For example, if the data used to train an ML algorithm are biased or unrepresentative, the resulting predictions may be inaccurate or unfair [44]. Ensuring the ethical and fair use of AI for the development of new therapeutic compounds is an important consideration that must be addressed [45]. Several strategies and approaches can be used to overcome the obstacles faced by AI in the context of chemical medicine. One approach is the use of data augmentation [46], which involves the generation of synthetic data to supplement existing datasets. This can increase the quantity and diversity of the data available for training ML algorithms, improving the accuracy and reliability of the results [47]. Another approach is the use of explainable AI (XAI) methods [48], which aim to provide interpretable and transparent explanations for the predictions made by ML algorithms. This can help to address concerns about bias and fairness in AI-based approaches [43] and provide a better understanding of the underlying mechanisms and assumptions behind the predictions [49].

Current AI-based approaches are not a substitute for traditional experimental methods, and they cannot replace the expertise and experience of human researchers [50,51]. AI can only provide predictions based on the data available, and the results must then be validated and interpreted by human researchers [52]. However, the integration of AI with traditional experimental methods can also enhance the drug discovery process [3]. By combining the predictive power of AI with the expertise and experience of human researchers [53], it is possible to optimize the drug discovery process and accelerate the development of new medications [54].

## 9. Ethical Considerations Regarding the Use of AI in the Pharmaceutical Industry

As discussed in the previous section, it is important to consider the ethical implications of using AI in this field [55,56]. One key issue is the potential for AI to be used to make decisions that affect people’s health and well-being, such as decisions about which drugs to develop, which clinical trials to conduct, and how to market and distribute drugs. Another key concern is the potential for bias in AI algorithms, which could result in unequal access to medical treatment and the unfair treatment of certain groups of people. This could undermine the principles of equality and justice. The use of AI in the pharmaceutical industry also raises concerns about job losses due to automation. It is important to consider the potential impact on workers and provide support for those who may be affected. Additionally, the use of AI in the pharmaceutical industry raises questions about data privacy and security. As AI systems rely on large amounts of data to function, there is a risk that sensitive personal information could be accessed or misused. This could have serious consequences for individuals, as well as for the reputation of the companies involved. The collection and use of sensitive medical data must be performed in a way that respects the individuals’ privacy and complies with the relevant regulations.

Overall, the ethical use of AI in the pharmaceutical industry requires careful consideration and the adoption of thoughtful approaches to addressing these concerns. This can include measures such as ensuring that AI systems are trained on diverse and representative data, regularly reviewing and auditing AI systems for bias, and implementing strong data privacy and security protocols. By addressing these issues, the pharmaceutical industry can use AI in a responsible and ethical manner.

## 10. Conclusions and Summary of the Potential of AI for Revolutionizing Drug Discovery

In conclusion, AI has the potential to revolutionize the drug discovery process, offering improved efficiency and accuracy, accelerated drug development, and the capacity for the development of more effective and personalized treatments (Figure 1). However, the successful application of AI in drug discovery is dependent on the availability of high-quality data, the addressing of ethical concerns, and the recognition of the limitations of AI-based approaches.

Recent developments in AI, including the use of data augmentation, explainable AI, and the integration of AI with traditional experimental methods, offer promising strategies for overcoming the challenges and limitations of AI in the context of drug discovery. The growing levels of interest and attention from researchers, pharmaceutical companies, and regulatory agencies, combined with the potential benefits of AI, make this an exciting and promising area of research, with the potential to transform the drug discovery process.

## 11. Expert Opinions from the Human Authors about ChatGPT and AI-Based Tools for Scientific Writing

As discussed in previous sections, AI has the potential to play a crucial role in the various stages of drug discovery, ranging from drug design to the final market introduction (Figure 1). However, the impact of AI extends beyond these areas and can greatly benefit the processing and analysis of scientific literature. Integrating AI into a literature review and article writing in the field of drug design holds immense promise. AI algorithms can expedite the review process, provide comprehensive insights from diverse data sources, and assist in identifying new research avenues. Additionally, AI-powered writing tools can enhance the quality and efficiency of scientific writing, enabling researchers to effectively communicate their findings. By embracing AI technologies in these domains, we not only save time and resources but also enhance the overall quality of drug design research, bringing us closer to the development of innovative and life-changing therapies.

The utilization of AI when composing literature reviews makes a significant contribution to drug discovery, which is the primary focus of this manuscript. Incorporating AI in the creation and evaluation of scientific literature offers several advantages that expedite the process of developing and approving new drugs. These advantages include efficient analysis of a large volume of articles, accurate extraction and summarization of information, access to up-to-date knowledge, the discovery of hidden insights, and integration across different disciplines. In the near future, the autonomous AI-assisted preparation of reviews is expected to become an integral part of the workflow of AI-assisted drug discovery. In the present review, we aimed to test the state-of-the-art AI tools for writing and revising the literature, contributing to the development of this research direction by establishing the initial foundations.

ChatGPT, a chatbot based on the GPT-3.5 language model (at the moment of preparation of this manuscript), has not been designed as an assistant for writing scientific papers. However, its ability to engage in coherent conversations with humans and provide new information on a wide range of topics, as well as its ability to correct and even generate pieces of computational code, has been a surprise to the scientific community. Therefore, we decided to test its potential to contribute to the preparation of a short review on the role of AI algorithms in drug discovery. As an AI assistant to write scientific papers, ChatGPT has several advantages, including its capacity to generate and optimize text quickly, as well as its ability to help users with several tasks, including the organization of information or even connecting ideas in some cases. However, this tool is in no way ideal as a technique to generate new content. Our revision of the text, which was generated by the AI following our instructions, required the application of major edits and corrections, including the replacement of nearly all the references since those provided by the software were clearly incorrect. This is a huge problem with ChatGPT, and it represents a key difference with respect to other computational tools such as typical web browsers, which are focused on providing reliable references for the required information. Another important problem of the employed AI-based tool is that it was trained in 2021 and so does not work with updated information. Most of these problems could be solved relatively quickly, which introduces new and urgent challenges regarding the control of apparently new content.

As a result of this experiment, we can state that ChatGPT is not a useful tool for writing reliable scientific texts without substantial human intervention. The program lacks the knowledge and expertise necessary to accurately and adequately convey complex scientific concepts and information. Additionally, the language and style used by ChatGPT may not be suitable for academic writing. In order to produce high-quality scientific texts, human input and human review are essential. One of the main reasons why this AI is not yet ready to be used in the production of scientific articles is its lack of ability to evaluate the veracity and reliability of the information that it processes. As a result, scientific text generated by ChatGPT will definitely contain errors or misleading information. It is also important to note that reviewers may find it not a trivial matter to distinguish between an article written by a human or by this AI. This makes it essential for the review processes to be thorough in order to prevent the publication of false or misleading information. A real risk is that predatory journals may exploit the opportunity for the quick production of scientific articles to generate large amounts of low-quality content. These journals, often motivated by profit rather than a commitment to scientific advancement, may use AI to rapidly produce articles, flooding the market with subpar research that undermines the credibility of the scientific community. One of the biggest dangers is the potential for the proliferation of false information in scientific articles, which could lead to a devaluation of the scientific enterprise itself. Losing trust in the accuracy and integrity of scientific research could have a detrimental impact on the progress of science.

There are several possible solutions for mitigating the risks associated with the use of AI in the production of scientific articles. One solution is to develop AI algorithms that are specifically designed for the production of scientific articles. These algorithms could be trained on large datasets of high-quality, peer-reviewed research, which would help to ensure the veracity of the information they generate. Additionally, these algorithms could be programmed to flag potentially problematic information, such as references to unreliable sources, which would alert researchers to the need for further review and verification. Another approach would be to develop AI systems that are better able to evaluate the authenticity and reliability of the information they process. This could involve training the AI on large datasets of high-quality scientific articles, as well as using techniques such as cross-validation and peer review to ensure that the AI produces accurate and trustworthy results. Another potential solution is to establish stricter guidelines and regulations for the use of AI in scientific research. This could include requiring researchers to disclose when they have used AI in the production of their articles, and implementing review processes to ensure that the AI-generated content meets certain standards of quality and accuracy. Additionally, it could also include requirements for researchers to thoroughly review and verify the exactitude of any information generated using AI before it is published, as well as penalties for those who fail to do so. It may also be useful to educate the public about the limitations of AI and the potential dangers of relying on it for scientific information. This could help to prevent the spread of misinformation and ensure that the public is better able to distinguish between reliable and unreliable sources of scientific information. Funding agencies and academic institutions could also play a role in promoting the responsible use of AI in scientific research by providing training and resources to help researchers understand the limitations of the technology.

Overall, addressing the risks associated with the use of AI in the production of scientific articles will require a combination of technical solutions, regulatory frameworks, and public education. By implementing these measures, we can ensure that AI is used responsibly and effectively in the world of science. It is important for researchers and policymakers to carefully consider the potential dangers of using AI in scientific research and to take steps to mitigate these risks. Until AI can be trusted to produce reliable and accurate information, it should be used with caution in the world of science. It is essential to carefully evaluate the information provided by AI tools and to validate it using reliable sources.

## Figures and Tables

**Figure 1 pharmaceuticals-16-00891-f001:**
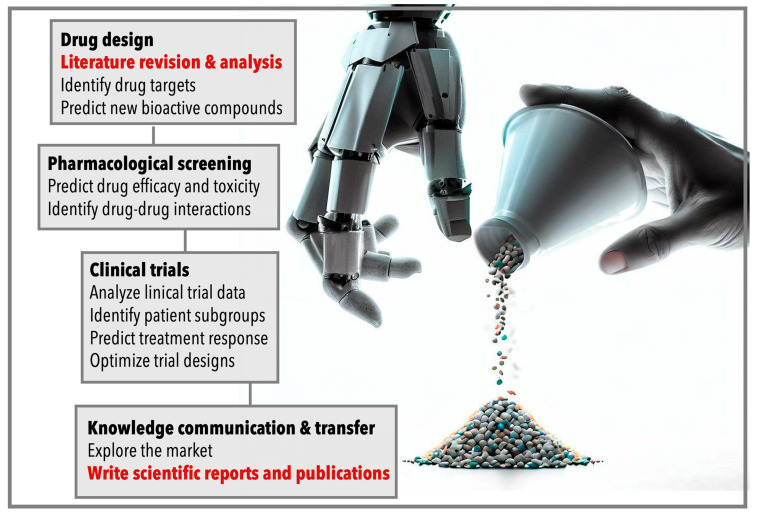
Graphical flowchart illustrating the development process of a pharmacologically active molecule, from design to knowledge communication and transfer. AI-based approaches complement traditional methods but still cannot replace human expertise. By combining the predictive power of AI with human researchers’ knowledge, the drug discovery process can be optimized and accelerated. The present work examines the cutting-edge advancements in the stages of “Literature revision and analysis” and “Write scientific reports and publications” (highlighted in red) using ChatGPT, a chatbot based on the GPT-3.5 language model.

## Data Availability

Data is contained within the article and Appendix A.

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
