# Peer review of "The Role of AI in Drug Discovery: Challenges, Opportunities, and Strategies"

_pharmaceuticals, 2023, doi:10.3390/ph16060891_

Round 1

Reviewer 1 Report

The authors in this manuscript used artificial intelligence ChatGPT to generate a text (i.e., manuscript) on the topic The Role of AI in Drug Discovery: Challenges, Opportunities, and Strategies. The authors iteratively corrected (supplemented) the text originally generated by AI. The authors also gave their opinion on the quality and applicability of the generated text.

I think the Author's main conclusion (with which I fully agree) is that AI can be applied to generate a review article if the AI learning set is exclusively a database of scientific publications.

Minor proposal:

The authors could themselves or with the help of AI generate a flowchart (Figure) of the development of a pharmacologically active molecule from structure planning (design) (QSAR), synthesis planning, synthesis, pharmacological screening to clinical trials and to state the importance of AI and the human factor at each step, i.e. at which stage AI has the greatest importance.

In the supporting material, the authors could provide a brief overview of the functioning of AI and the algorithms it is based on, some algorithms are also mentioned in the manuscript.

The review was done by a human being.

Reviewer 2 Report

This review article on the potential of AI in drug discovery serves as an interesting experiment to test the ability of ChatGPT, an AI language model, to assist human authors in writing review articles. It covers the benefits of AI, such as improved efficiency and cost-effectiveness, as well as challenges like data quality and ethical concerns. Specific applications of AI in drug discovery are explored, along with strategies for integrating AI with traditional methods. The article acknowledges limitations, such as the need for high-quality data, and emphasizes the responsible and ethical use of AI. Overall, it provides valuable insights for researchers interested in this field. 

Indeed, the authors' attempt to incorporate ChatGPT in the paper is an interesting and innovative approach. The use of AI technologies like ChatGPT in academic writing is a growing area of exploration that holds promise for generating new ideas and perspectives. By leveraging ChatGPT, the authors have demonstrated a willingness to explore novel methods of content creation and harness the capabilities of AI. However, it is important to balance this experimentation with the need for in-depth analysis and critical evaluation. While the attempt is commendable, it is essential to supplement the generated text with additional research, expert insights, and rigorous examination to ensure the paper's overall depth and scholarly rigor.

The paper appears to have a slight distraction in its focus as it simultaneously discusses two distinct aspects: the role of AI in drug discovery and the use of ChatGPT to generate the content for the review article itself. While both topics are intriguing, their inclusion within the same paper might create a lack of clarity and coherence in the overall narrative. It is recommended that the authors place a stronger emphasis on the role of AI in drug discovery within the paper. While the use of ChatGPT to generate the content is an interesting aspect, it should be treated as a supporting element rather than the primary focus. By shifting the focus towards the exploration of AI's potential in drug discovery, the authors can provide a more comprehensive and in-depth analysis of the topic.

Round 2

Reviewer 1 Report

The authors improved the manuscript and answered any doubts.

Author Response

Comments and Suggestions for Authors are: "The authors improved the manuscript and answered any doubts". Thanks a lot!

Reviewer 2 Report

Most of the concerns raised earlier have been appropriately addressed in the revised manuscript. I agree with the authors' viewpoint that integrating AI into the literature review in drug design should be considered an integral part of AI's application in drug discovery.

To further enhance clarity and transparency regarding the respective contributions of humans and ChatGPT in the manuscript, it is advisable for the authors to specify explicitly, within the supplementary information (SI), the specific sections or materials that humans added. As an example, ChatGPT solely suggested the work conducted by Gupta, R. et al. and the identification of novel inhibitors of beta-secretase (BACE1) in section 5, while the author complemented the content by including additional materials relating to the discovery of drugs for the treatment of COVID-19.

By addressing this aspect, the review will provide a comprehensive and well-rounded examination of AI's potential and limitations in the context of writing literature reviews for drug discovery.

Author Response

Dear Reviewer,

Thank you for your thoughtful review and helpful feedback on our manuscript. We greatly value your suggestion for increased clarity and transparency concerning the respective roles of human authors and AI in the preparation of this manuscript.

We would like to highlight that the detailed account of the interaction between the human authors and the AI is already included both in the main body of our current manuscript and the Supplementary Information (SI). The SI specifically includes eight pages outlining the exact inputs provided to ChatGPT (accessed on December 6th, 2022), as well as the corresponding outputs for each query. Additionally, our manuscript itself explicitly describes the process and the differences between the original ChatGPT version and the current one:

“For the purposes of this paper, the human-authors provided the input, including the topic of the paper (the use of AI in drug discovery), the number of sections to be considered, as well as the specific prompts and instructions for each section. The pieces of text generated by the AI were edited to correct and enrich the content, to avoid repetitions and inconsistences. All the references suggested by the AI were also revised. The final version of this work resulted from an iterative process of revisions by the human authors assisted by the AI. The total percentage of similarity between the preliminary text, obtained directly from ChatGPT, and the current version of the manuscript is: identical 4.3%, minor changes 13.3% and related meaning 16.3%2. The percentage of correct references in the preliminary text, obtained directly from ChatGPT, was just 6%. The original version generated by ChatGPT, along with the inputs used to create it, are included as Supporting Information.”

As you can see, the manuscript clearly presents the similarity between the preliminary text, obtained directly from ChatGPT, and the final version: 4.3% identical, 13.3% with minor changes, and 16.3% with related meaning. The low accuracy of references in the preliminary text, obtained directly from ChatGPT, was just 6%, and unfortunately, none of the original references provided by the AI were applicable.

To further assist your review, we are resubmitting our manuscript and the SI with the sections that elaborate on these points highlighted in green. This will help you easily locate the information that addresses the extensive degree of human intervention required to produce the final manuscript from the preliminary AI-generated version. We are confident that this will further clarify the integral roles of AI and human authors in creating this manuscript, in line with your constructive comments.

We appreciate your time and effort in reviewing our work and look forward to your continued feedback.

Sincerely,

Rebeca.